# Damping Optimization and Energy Absorption of Mechanical Metamaterials for Enhanced Vibration Control Applications: A Critical Review

**DOI:** 10.3390/polym17020237

**Published:** 2025-01-18

**Authors:** Salem Bashmal, Aamer Nazir, Sikandar Khan, Abdulrahman Alofi

**Affiliations:** 1Department of Mechanical Engineering, King Fahd University of Petroleum & Minerals, Dhahran 31261, Saudi Arabia; g202202260@kfupm.edu.sa (F.); bashmal@kfupm.edu.sa (S.B.); sikandarkhan@kfupm.edu.sa (S.K.); amaloufi@kfupm.edu.sa (A.A.); 2Interdisciplinary Research Center for Intelligent Manufacturing and Robotics, King Fahd University of Petroleum & Minerals, Dhahran 31261, Saudi Arabia; 3Interdisciplinary Research Center on Advanced Materials, King Fahd University of Petroleum & Minerals, Dhahran 31261, Saudi Arabia

**Keywords:** polymers, additive manufacturing, mechanical metamaterials, lattice structures, vibration and damping, bandgap formation, energy absorption

## Abstract

Metamaterials are pushing the limits of traditional materials and are fascinating frontiers in scientific innovation. Mechanical metamaterials (MMs) are a category of metamaterials that display properties and performances that cannot be realized in conventional materials. Exploring the mechanical properties and various aspects of vibration and damping control is becoming a crucial research area. Their geometries have intricate features inspired by nature, which make them challenging to model and fabricate. The fabrication of MMs has become possible because of the emergence of additive manufacturing (AM) technology. Mechanical vibrations in engineering applications are common and depend on inertia, stiffness, damping, and external excitation. Vibration and damping control are important aspects of MM in vibrational environments and need to be enhanced and explored. This comprehensive review covers different vibration and damping control aspects of MMs fabricated using polymers and other engineering materials. Different morphological configurations of MMs are critically reviewed, covering crucial vibration aspects, including bandgap formation, energy absorption, and damping control to suppress, attenuate, isolate, and absorb vibrations. Bandgap formation using different MM configurations is presented and reviewed. Furthermore, studies on the energy dissipation and absorption of MMs are briefly discussed. In addition, the vibration damping of various lattice structures is reviewed along with their analytical modeling and experimental measurements. Finally, possible research gaps are highlighted, and a general systematic procedure to address these areas is suggested for future research. This review paper may lay a foundation for young researchers intending to start and pursue research on additive-manufactured MM lattice structures for vibration control applications.

## 1. Introduction

Recent advances in engineering materials depend only on changing the composition of a material to alter its quality. To satisfy this demand, materials are customized to have specific qualities [1]. However, because materials naturally have certain intrinsic features, materials with lower strength have lower density, and vice versa. Producing materials with specific intrinsic properties is challenging [2]. By creating an optimal architecture, natural materials, particularly cellular materials, display intriguing features that are absent in conventional materials [3]. Many topological designs that shrink the size from a larger scale to a smaller unit-cell scale to provide the same strength and qualities have been presented in an attempt to reduce the connection between mass density and mechanical properties. Advancements in AM, commonly known as three-dimensional (3D) printing, have made it possible to create complex architectures for a variety of polymers and other engineering materials with nanometer-length scales [4]. AM uses a layer-by-layer addition of materials to fabricate intricate features and shapes [5,6,7]. Several AM techniques, including stereolithography (SLA), fused deposition modeling (FDM), selective laser sintering (SLS), and selective laser melting (SLM), have been used to precisely and accurately create complex mechanical metamaterial (MM) structures [8,9,10,11,12,13,14,15,16,17].

A noteworthy area of scientific discovery, metamaterials, is expanding the limits of conventional materials and bringing about a new era of possibilities never seen before [18]. They have various functions and peculiar characteristics [19], such as negative permittivity, negative permeability, negative stiffness, a negative bulk modulus, thermal expansion coefficient sign reversal, a negative mass density, a negative refractive index, and a negative Poisson’s ratio [20,21]. Metamaterials have accelerated technological advancements in several fields. For example, their application in optics has advanced the development of superlenses [22,23,24], allowing for high-resolution imaging. Metamaterials have significantly improved the transmission and reception of signals in telecommunications, helping to develop structured waveguides [25,26] and antennas [27]. Additional applications of the metamaterial idea include solid mechanics [28], elastic waves [29], acoustics [30,31], and vibration absorption. Research on the vibrational isolation of metamaterials was initiated [32] after studying locally resonant phononic crystals. Metamaterials can be classified as optical, acoustic, electromagnetic, or mechanical, as shown in Figure 1.

Several studies have investigated the mechanical properties of various MM lattices. The compressive properties of different cell topologies, including body-centered cubic (BCC), BCCZ (BCC with z-axis reinforcement) [33], 3D re-entrant lattice [34], gyroid structure [35], tetrakaidecahedron, diamond, BCC [36], BCC, BCCZ, face-centered cubic (FCC), FCC with vertical struts (FCCZ) [37], BCC, BCCZ, FCC, FCCZ, FCC and BCC with z-strut (FBCCZ) [38] topologies, fabricated using the SLM AM technique have been studied. The focus of past research has been to study their compressive mechanical properties. However, the vibration and damping control aspects of these polymers and other engineered materials have been studied to a limited degree and must be explored in depth for engineering applications.

In this review, the focus is on recent research on various aspects of the vibration and damping control of MMs fabricated using polymers and other engineering materials, which may lay a starting point for researchers interested in exploring this domain. Section 2 covers the detailed literature on MMs manufactured using polymers and other engineering materials for various aspects of vibration control, including bandgap formation, energy absorption, and damping control. In Section 3, the vibration-damping analytical modeling and experimental measurement of MMs are presented. Section 4 suggests the possible research gaps and future systematic research procedures. Finally, Section 5 concludes this paper.

## 2. Mechanical Metamaterial for Vibration Control

Mechanical metamaterials exhibit mechanical properties that are not found in conventional materials, that is, negative stiffness (NS), a negative Poisson’s ratio, a negative mass density, and a negative bulk modulus, as highlighted in Figure 1 [39,40,41,42]. These materials have attracted the interest of researchers since 2010 [43] owing to their unique mechanical properties. Because of these features, MMs can be used in a variety of engineering applications, including bandgap formation, energy absorption, mechanical energy dissipation [42,44,45,46], seismic safekeeping [47], and vibration damping control [48,49,50]. Vibration is the process in which a mechanical structure undergoes back-and-forth motion about an equilibrium position when subjected to an external force disturbance [51]. Vibrations can be found in any component that is elastic and has inertia; thus, most mechanical structures experience vibrations [52]. These are undesirable because they can cause fatigue and successive machinery failure, resulting in human discomfort and fatal failures. Vibration examples include running heavy-duty machinery in industries, civil bridges due to vehicle movement, and tall-structure vibrations due to winds. These unfavorable vibrational effects have motivated researchers to develop various materials, engineering systems, and plans to mitigate, suppress, and absorb undesirable vibrations [53].

MMs may be used in dynamic and vibration environments to form frequency bandgaps and absorb and damp undesirable vibrational energy through various mechanisms, including internal friction, structural reconfiguration, and mode conversion. The geometric design of MM lattice structures can be modified and optimized, affecting the natural frequency of the system and leading to bandgap formation and the damping of vibration amplitudes, thereby attenuating and suppressing the overall vibration of the system. The vibration control aspects of MMs can be categorized into two types based on the literature covered in this study. The first category involves bandgap formation and energy absorption using MMs. The second category comprises damping control. Both categories aim to suppress, attenuate, isolate, absorb, and dissipate the vibrations. Bandgap formation focuses on stopping the desired frequencies from passing through structures using locally resonating structures, thereby avoiding vibration generation. Energy absorption from vibrations, shocks, and impacts of the structures may be enhanced using NS and a negative Poisson’s ratio structural configuration [54]. The damping of a structure can be changed by varying the natural frequency and damping constants of the system. Figure 2 categorizes the vibration control of an MM into bandgap formation, energy absorption, and damping control. Figure 2 shows some of the different MM configurations fabricated from polymers and other engineering materials reported in the literature to achieve vibration control.

### 2.1. Bandgap Formation and Energy Absorption Using Mechanical Metamaterials

Bandgap formation refers to the stopping of certain undesirable vibration frequencies from passing through a structure. MMs are formed by assembling subunits in the parent building structure so that mechanical waves or vibrations passing through the parent structure resonate with the subunits. The phenomenon from local resonance (LR) results in the creation of a phononic bandgap, referred to as a stopband. Mechanical waves cannot pass through this stopband of the parent building structure within a given frequency range [62]. Brag scattering and light bending are two approaches used to study stopband characteristics. The principles behind these two techniques for mechanical wave attenuation are discussed in [63]. The LR approach was also used to form stopbands in low-frequency regions. The LR process produces a low-frequency bandgap with a lattice constant that is much lower than the wavelength of the transmitting mechanical waves. The lattice constant refers to the repeating unit length of the metamaterial’s structure. The bandgap with a bandwidth event produced by the LR process is a function of the geometric and material characteristics of the resonators and does not depend on the repetition or unit cell arrangement [64].

Several studies have been conducted on different MM configurations to achieve a low-frequency bandgap formation for vibration control and energy harvesting. El-Borgi et al. [55] used metamaterial beams in their numerical and experimental studies, as shown in Figure 3a. They observed two bandgaps where the vibration was attenuated. They investigated a configuration with a fixed number of resonators. This study can be extended by using different numbers of resonators with different point masses at the ends. Anigbogu et al. [56] used periodic structures with local resonators to attenuate vibration and harvest energy, as shown in Figure 3b. Their study suggested several design considerations, including the dimensions and tilt angle of the cantilever beams, number of unit cells, and size and position of the central magnets, to enhance the overall performance of manufactured metamaterial structures with dual functions (vibration attenuation and energy harvesting). They mentioned that it is necessary to develop a thorough and sophisticated coupled model of the magneto-mechanical metamaterial structure capable of energy harvesting from vibration attenuation. Similarly, star-shaped unit cells [65] were studied for stopband improvement and bandwidth widening in varying frequency ranges. They mentioned that the use of metamaterials for controlling vibrations requires a complete study of their fatigue behavior. The authors also mentioned that progress in design and software simulations will allow the creation of new designs, modeling, and topology optimization using artificial intelligence (AI) methods. AI methods, such as generative design and topology optimization, are used to design and optimize novel MM lattice structures for improved vibration control.

An L-joint LR beam [66] was studied for low-frequency vibration attenuation in engineering applications. They found that the longitudinal force-moment resonators suppressed the transmission of axial waves in the low-frequency range in the third and fourth cells in the LR beam. They specified that the proposed investigation could further facilitate future work, including health monitoring and damage detection. In addition, applying the proposed technique to the study of the bandgap characteristics of space LR frames is a fascinating topic for future research. Ji et al. [67] reviewed metamaterials and origami structures for vibration mitigation and isolation. The authors highlighted and suggested important considerations, including nonlinear study, topology optimization, and 3D optimum designs for nonlinear structures in broad applications to control vibration and sound. In addition, expanding the research on the mechanical performance of MMs to nonlinear and 3D studies is expected to produce encouraging scientific outcomes in the future. This is expected to find stiffness attributes in three directions and, hence, offers encouraging structures for designing new vibration isolators to avoid multidirectional vibrations transmitted to environmental objects. In addition, the auxetic properties of the dual stable states and origami-based structures have great potential for several dynamic applications, including energy absorption and dissipation. Finally, they highlighted that real examples of origami-based structures for vibration isolation are lacking in real-world applications. Auxetic metamaterials are mechanical metamaterials that exhibit negative Poisson ratios. They possess excellent energy absorption and vibration-damping capabilities for seismic and automotive applications [68,69,70].

A layered metamaterial structure [57] was used for frequency bandgap formation, as shown in Figure 3c. The results concluded that the low-frequency bandgap could be further decreased by designing local resonators. In addition, altering the length of the local resonator primarily affects the shape and location of the local resonator frequency bandgap. An internally coupled metamaterial beam [71] was used for low-frequency energy harvesting and vibration suppression. It was found that although internal coupling with such a beam connection does not result in a significant improvement in energy harvesting compared with conventional metamaterial beam piezoelectric energy harvesters, the vibration suppression capability is significantly improved with a wider second bandgap. A pyramidal-truss metamaterial beam [72] was investigated for low-frequency vibration insulation. It was observed that the position and bandgap width could be chosen by adjusting the natural frequency of the resonators, the truss inclination angle, and the mass ratio of the face sheets and resonators. Similar studies on bandgap formation for vibration control have been conducted, along with review articles, and are summarized in Table 1. Additionally, the energy dissipation and absorption of the metamaterial structures were investigated, as shown in Figure 3 and Table 1. Some examples of energy dissipation include polymeric architectured unit cells with a three-spring configuration model for energy dissipation [40], as shown in Figure 3f, and NS honeycombs that use elastic-buckling phenomena for energy dissipation [73]. In addition, energy absorption examples include a cubic NS lattice structure [74]; a snap-fit mechanical metamaterial [61], as shown in Figure 3h; and a novel ZBSO metamaterial [60], as shown in Figure 3g. Similarly, Timoshenko beams with periodic two-degrees-of-freedom (DOF) uncoupled force-moment-type resonators [58] and HSLDS resonators [59] were studied for low-frequency bandgap formation, as shown in Figure 3d,e. Similar studies on energy dissipation and absorption for vibration control have been conducted and are summarized in Table 1.

Several noteworthy limitations were present in some research that analyzed MMs for vibration control and energy harvesting. A prevalent problem is the limited range of parametric variations, as demonstrated by studies such as those by El-Borgi et al. [55] and Anigbogu et al. [56], which concentrated on specific resonator arrangements without delving into more general design variables, such as changing the number of resonators, altering the geometry, or changing the mass distribution. Furthermore, although these investigations demonstrated the possibility of low-frequency bandgap creation, Ji et al. [67] pointed out that they mostly depended on oversimplified linear models and ignored the crucial role of nonlinear dynamics. The lack of this element significantly diminishes the suitability of these systems in actual dynamic settings. Furthermore, many studies, such as those examining multilayer MMs and star-shaped unit cells, ignore structural durability and fatigue behavior, particularly under long-term cyclic loading. Additionally, there is a lack of thorough experimental validation, with many studies being strongly dependent on numerical models, especially for more intricate metamaterial systems that are mostly theoretical, such as origami-based structures. Moreover, the research discusses but does not completely incorporate advanced optimization approaches, such as topology optimization and AI-driven design, which limit the possibility of producing creative and reliable solutions. These restrictions reveal several important research needs, including the use of advanced optimization techniques, comprehensive experimental validation to verify theoretical models, the study of nonlinear dynamics in vibration control, and the study of structural durability and fatigue behavior.

### 2.2. Damping Control Using Mechanical Metamaterials

Vibration damping refers to the loss of undesirable vibrational energy from a vibrating system [95]. Because of the removal of energy, the vibration amplitude and duration of the system are reduced. There are three categories of vibration damping: viscous, Coulomb, and hysteresis. Viscous damping dissipates the vibration energy of a system using a viscous fluid medium. This method is a common type of damping and is easily incorporated into mathematical models. Coulomb damping refers to energy dissipation due to dry friction between two surfaces in contact. The last category is hysteresis or structural damping, which results from internal energy loss within a system [95,96].

Subsequently, various studies focusing on damping control in different MM lattice structures will be reviewed. Monkova et al. [10] studied the mechanical vibration damping and compression properties of a polymeric BCC lattice structure, as shown in Figure 4a. They assessed testing samples subjected to harmonic excitation at three attached inertial masses and experimentally demonstrated that the damping performance increased with decreasing volume ratio and increasing cell size, inertial mass, and excitation frequency. In another study, the dynamic response of lattice structures (FBCCZ lattice structure) in medium-to-high-frequency intervals was studied experimentally and modeled, as shown in Figure 4b. They experimentally showed that lattice structures have excellent damping properties compared with solid materials with equivalent static stiffness [11]. Wang et al. [12] investigated the effect of viscoelastic material filling (VMF) on the structural and vibrational performance of a lattice truss, as shown in Figure 4d. They showed experimentally that the VMF method is effective in decreasing the vibration magnitude and has the potential for bandgap design.

In another study, the effects of the experimental arrangement conditions on various SLM AISI 316 L lattice structures were analyzed, as shown in Figure 4c. The dynamic response was found with the help of pulse testing and sinusoidal excitation using an electromagnetic shaker. Material damping was accurately calculated using noncontact sensors and specified boundary conditions [13]. Zhang et al. [14] studied the I-wrapped package (IWP-type) triply periodic minimal surface (TPMS) lattice structures and BCC-matching parts with the same topology, as shown in Figure 4e. Their results showed that the stiffness and frequency of the structure were directly related to the volume fraction and inversely related to the cell size. In another study, a new metamaterial comprising unit cells of rectangular sheets with conical springs was proposed, as shown in Figure 4f. They observed high transmission losses in the low-frequency range. They also studied the effects of the geometric parameters and found that they affect low-bandgap control [53].

In another study, a novel three-dimensional lightweight lattice structure with acoustic black hole (ABH) characteristics was proposed, as shown in Figure 4g. Their results showed that the proposed SC-BCC-ABH had superior stiffness compared with other lattice structures. In terms of vibration attenuation, the proposed lattice structure produced a wide pseudo-bandgap [15]. In another study, a polymeric MM with NS characteristics was fabricated to achieve high energy loss and low-frequency vibration suppression. Their results showed that the proposed metamaterial exhibited excellent vibration attenuation performance in the low-frequency range [16]. In another study, a technique to include tuned mass dampers (TMDs) in a lightweight, optimized structure was presented, as shown in Figure 4h. They placed multiple polymeric TMDs inside unit cells. They applied this technique to a two-segment robot arm, resulting in a 3.6% lighter structure, maintaining the same performance and conveying a 60% smaller dynamic displacement in a given frequency range [17]. In another study, the dynamic performance of gyroid structures was investigated. They studied the effects of the geometric parameters of the gyroid structure on dynamic characteristics. Their simulation results showed that the dynamic performance of the presented structure could be further increased using parametric models considering the frequency response [97].

The previously mentioned research is summarized in the following Table 2.

Several restrictions on MMs for damping control research were reviewed. Although Monkova et al. [10] and others showed how variables such as volume ratio, cell size, and inertial mass affect vibration damping, these studies mainly concentrated on geometric configurations and materials (such as ABS, AlSi10Mg, or Kagome trusses) without investigating a wider variety of materials and lattice topologies. Furthermore, several studies, including those by Zhang et al. [14] and Wang et al. [12], were limited to linear dynamic responses and ignored nonlinear behaviors that can be crucial for practical applications under challenging loading circumstances. Moreover, although some studies have examined filling materials, such as viscoelastic materials, little is known about how various filling types interact with structural damping. Finally, the simplicity of the excitation modes and boundary conditions used in the experimental settings restrict the applicability of the findings to complex real-world scenarios. These limitations highlight several important research requirements.

## 3. Vibration Damping Modeling and Experimental Measurement

An MM can be modeled as a single degree of freedom using a simple spring–mass–damper arrangement, as shown in Figure 5. Therefore, the vibration performance depends on the mass, stiffness, and damping constant of the structure. Figure 5 shows the behavior of the structure with and without damping when subjected to a range of forcing frequencies. In the case of a low damping value of the structure, the magnitude of the vibration amplitude ratio was high, and vice versa.

A single-DOF system subjected to harmonic excitation with damping is modeled using the following equation:(1)$$m\ddot{x}+c\dot{x}+kx=F_{o}coswt$$
where *m* is the system inertia, *c* is the damping constant, *k* is the stiffness, $F_{o}$ is the external force magnitude, and w is the excitation frequency.

Researchers have used displacement transmissibility equations for a structure. The equation used is as follows [98,99,100,101]:(2)$$T_{d}=\frac{x_{0}}{x_{1}}=\sqrt{\frac{k^{2}+{\left(c\omega \right)}^{2}}{{\left(k-m\omega^{2}\right)}^{2}+{\left(c\omega \right)}^{2}}}=\sqrt{\frac{1+{\left(2\xi r\right)}^{2}}{{\left(1-r^{2}\right)}^{2}+{\left(2\xi r\right)}^{2}}}$$
where the damping ratio ξ and frequency ratio *r* are given by the following expressions [102,103]:$$\begin{array}{c} \xi =\frac{c}{2\sqrt{km}}=\frac{c}{2mw_{n}}\\ r=\frac{\omega }{\omega_{n}}=\frac{w}{\sqrt{\frac{k}{m}}} \end{array}$$

Scalzo et al. [11] used another set of formulations described below. They assumed a cantilever beam lattice as a single harmonic oscillator to characterize its modal parameters.(3)$$W\left(j\omega \right)=\frac{1}{m{\left(j\omega \right)}^{2}+c\left(j\omega \right)+K}=\frac{G}{{\left(\frac{j\omega }{\omega_{n,1}}\right)}^{2}+2\xi \left(\frac{j\omega }{\omega_{n,1}}\right)+1}$$
where *m*, *c*, and *k* are the modal mass, damping, and stiffness coefficients, respectively; *G* is the static compliance; and *W*(*jω*) is the dynamic compliance, while$$\omega_{n,1}=\sqrt{\frac{k}{m}},\;\;\;\xi =\frac{c}{2\sqrt{km}}$$ are defined as the natural pulsation/frequency and the damping ratio, respectively.

They used the classical Rayleigh model to determine the damping behavior of the testing samples, and using this model in combination with Equation (3) at resonance resulted in the desired damping ratio, expressed as follows:(4)$$\omega_{n}≅\omega_{r}\left[\frac{rad}{sec}\right],\;\;\xi ≅\frac{G}{2\left|W\left(j\omega_{r}\right)\right|}$$

Another widely used technique is half-power bandwidth, which measures the damping ratio of a structure [96,104,105,106]. In this technique, a structure is excited by forced vibrations using a harmonic force. First, the amplitude of the vibration response is calculated in the frequency domain. Subsequently, the resonant vibration amplitude Q of the structure is measured at its natural frequency ($w_{n})$. The respective frequencies of the system at $\frac{Q}{\sqrt{2}}$ or −3 dB ($w_{2}\;\mathrm{and}\;w_{1})$ from the peaks of the vibration amplitude are then calculated. The damping ratio can be calculated from the modal loss factor $\eta_{k}$ using the following equations [103]:(5)$$\eta_{k}=\frac{w_{2}-w_{1}}{w_{n}}$$(6)$$\eta_{k}=\frac{2\xi_{k}}{\sqrt{1-\xi_{k}^{2}}}⋍2\xi_{k}$$

Another simple method, known as the logarithmic decrement method, is used to measure the damping ratio of a structure [96,106,107,108,109]. In this method, the damping ratio is measured experimentally by exciting the structure using an initial condition, that is, the displacement or velocity, and the decay of the free response is recorded [110]. This is also known as free vibration, and there is no force excitation. The damping ratio of the structure is calculated using the following equation:(7)$$\xi =\frac{\delta_{l}}{\sqrt{{\left(2\pi \right)}^{2}+{\delta_{l}}^{2}}}$$
where $\delta_{l}$ is the logarithmic decrement measured from the decay curve of the response.

The experimental facility used by different researchers to measure the dynamic characterization of mechanical metamaterial lattice structures differs from the arrangement used to test the specimens. The research uses different techniques and experimental facilities, as shown in Figure 6. The testing samples are in the form of a beam shape excited by an impact hammer or shaker. The specifications of Figure 6a are as follows: The dynamic response of the specimen was evaluated using experimental modal analysis. Each specimen was secured using a vice on a Haas VF-2TR CNC milling machine rotating table. The free tip of the testing sample was hit using a Dytran type 5800B4 (sensitivity 2.41 mV/N) linked to a Kistler amplifier type 5134 B. An eddy NCDT3010-M controller was linked to an eddy current probe Micro-Epsilon type ES1 (range: approximately ±0.5 mm) to monitor the ensuing vibrations. To confirm that the dynamic compliance of the specimen was greater than that of the vice, a triaxial accelerometer Kistler type 8763B (sensitivity: 50 mV/g) was placed on the vice close to the specimen. A National Instruments cDAQ-9178 device with NI9215 modules connected to a PC via a USB was used to acquire the signals. A sampling rate of 20 kHz was used. A MathWorks MATLAB environment was used to execute the data elaboration process [11]. Figure 6b shows photographs of the polymeric metastructures excited by a shaker. A computer equipped with the SCO-107 sine sweep software package was used to generate the driving signals. This signal was sent to an LDS V408, 10/32UNF vibration shaker after being sent to an LAS200 controller and an LPA100 amplifier. The amplitude and frequency of the drive signal were managed using a lower accelerometer (4534-B) with a sensitivity of 10 mV/g. For feedback, the signal from the accelerometer was also sent to the controller. A second accelerometer with the same specification of 10mV/g was mounted on top of the metastructure to obtain the response in the vertical direction owing to longitudinal vibrations [111].

Figure 6c shows the mechanical metamaterial structure with a base support attached to the shaker. A signal generator (Agilent 33220A, Santa Clara, CA, USA) was used to generate a sine sweep excitation from 50 to 2000 Hz. A power amplifier was used to amplify the excitation signal (LDS PA100E). Sensors 1 (TLD333B30: 101.3 mV/g) and 2 (TLD352C03: 10.12 mV/g) were used to monitor the input excitation and output response signals, respectively. Dynamic signal acquisition with a sampling frequency of 6400 Hz and an analysis system (m + p VibPilot) were used to gather the data [15]. Figure 6d shows the facility used for testing a polymeric cantilever beam manufactured from a mechanical metamaterial. A force sensor was used to track the input force, an accelerometer was used to measure the amplitude of the vibration response, a dynamic electroshaker was used with a signal power amplifier, and a signal generator was used to excite the mechanical metamaterial. Mild-steel test equipment was employed to prevent undesired ambient vibrations from affecting the system. Data Physics Quattro Ace was used in the experiment to generate random excitation with a frequency range of 0–1200 Hz, which was then amplified by a signal power amplifier. Next, the Data Physics Signal Calc 240 Dynamic Signal Analyzer software was used to evaluate the measured data [112].

After reviewing the experimental facility in the literature, it was observed that the necessary equipment for calculating the modal parameters of an MM, including the natural frequency, mode shape, and damping coefficients, comprises a signal generator, amplifier, and vibration shaker or hammer. In addition, a highly sensitive and high-resolution accelerometer is required to record the vibration amplitude. A data acquisition system is used to filter and amplify the vibration signal data received from the accelerometer. The collected data must be processed and analyzed on a computer using a data-acquisition system-compatible and supported software. Figure 7 summarizes the overall layout of the basic experimental facility for the vibration-damping testing of an MM.

## 4. Possible Research Gaps and Future Research Systematic Procedure

Herein, research on mechanical metamaterials has been presented, covering bandgap formation, energy dissipation, energy absorption, and damping optimization for vibration control in different engineering applications. Several MM lattice structures have been investigated for damping enhancement, including the BCC, FBCCZ, Kagome lattice, NS, and gyroid structures. From the literature covered carefully in this paper, the following are some of the possible research gaps that may be addressed in the future to fully explore the vibration and damping control of MMs. Because it mostly focuses on MM lattice structures with one or a few characteristics, such as volume ratio and relative density, the research under discussion has a substantial gap in its scope. This method ignores the crucial impact of various variables on the vibration- and damping-controlling capabilities of an MM, including its structural shape, size, angle, orientation, and relative density. A more comprehensive investigation should be conducted to fully assess these criteria. In addition, the range of existing lattice structural morphologies has been largely overlooked in the current literature. In particular, there are noteworthy unknown morphologies based on the plate, strut, and surface configurations that may provide important insights. Furthermore, the numerical analysis lacks depth, focusing only on one lattice structure before proceeding to experimental validation. To improve the design of MM lattice structures for optimal vibration and damping performances, a more thorough parametric numerical analysis is required. Subsequently, the accuracy and applicability of the numerical predictions should be confirmed through experimental validation of the improved structure.

Another focus area is the practical application of MM lattice structures, which is the subject of this study. The practical performance and problems of the MM structures under examination are not well understood because they have rarely been tested in real-world engineering contexts. To overcome this problem, it is essential to use these lattice structures in authentic engineering settings where their efficacy and possible drawbacks can be assessed in real-world operating scenarios. Moreover, there is still much to learn about the possibilities for hybrid lattice architectures with different relative densities and gradients to improve vibration and damping performance. The study of these hybrid structures may lead to important developments in vibration-damping technology. Similarly, a deeper investigation into multilayer material modeling is required, particularly regarding the fusion of soft and rigid materials. Future designs may be more successful because of these studies, which improve the understanding of how various material combinations affect vibration damping.

A detailed systematic schematic is proposed in Figure 8, keeping in mind the above recommendations for young researchers to conduct in-depth investigations in the future to explore the vibration and damping of MM lattice structures.

## 5. Conclusions

In recent years, there has been an increase in research on MMs manufactured from polymers and other engineering materials for vibration and damping control applications. From the literature reviewed in this paper, it is evident that researchers are in the early stages of investigating and exploring the vibration and damping control of MMs. Therefore, a comprehensive review covering important vibration aspects, including bandgap formation, energy absorption, and damping control, was presented. Moreover, the respective research limitations with possible gaps were highlighted, including the use of a limited range of parametric variations, the use of oversimplified linear models, the ignoring of the crucial role of nonlinear dynamics, the ignoring of structural durability and fatigue behavior, the lack of thorough experimental validation for intricate origami-based structures, the need to incorporate advanced optimization approaches, such as topology optimization and AI-driven design, the need to use a wider variety of materials and lattice topologies, and the need to understand how various filling types interact with structural damping and incorporate real-world scenario excitation modes and boundary conditions rather than simplified ones. In addition, the vibration-damping modeling of an MM and its experimental measurements, along with different test facilities, were discussed. Furthermore, a general experimental facility layout was suggested for the experiment. Finally, the possible research gaps, including the need for a detailed morphology and parameter (size, angle, orientation, and relative density) study, thorough parametric numerical analysis, hybrid lattice architectures (with different relative densities and gradients), and assessment in real-world operating scenarios of MMs were critically highlighted. Finally, a systematic procedure for future research was proposed, including a step-by-step methodology for designing the best-performing MM structure for researchers interested in exploring this area. This review aims to establish a foundational framework for emerging researchers seeking to initiate and advance their studies on additively manufactured MMs for vibration control applications.

## Figures and Tables

**Figure 1 polymers-17-00237-f001:**
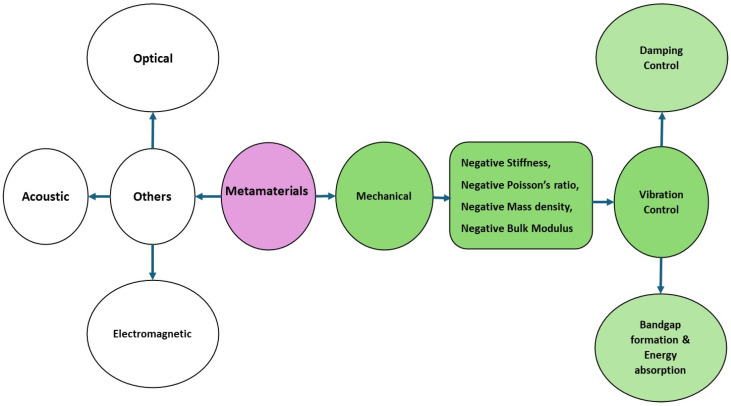
Classification of metamaterials with MMs for vibration control.

**Figure 2 polymers-17-00237-f002:**
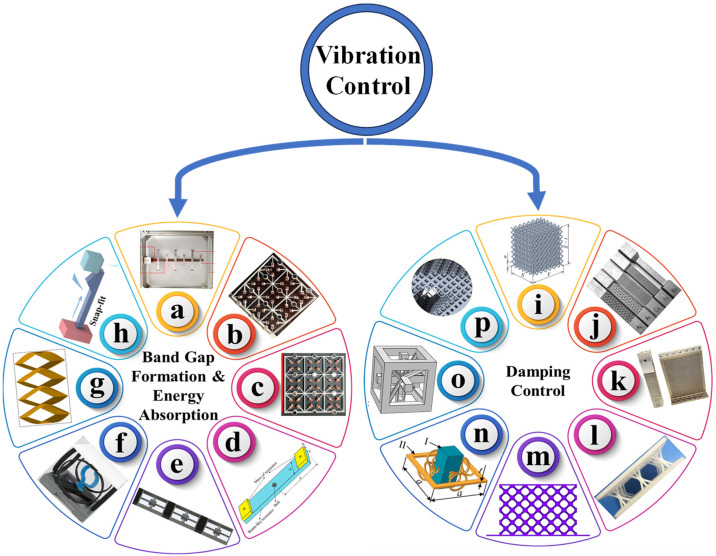
Several MM structures from the literature on vibration control: (**a**) [55], (**b**) [56], (**c**) [57], (**d**) [58], (**e**) [59], (**f**) [40], (**g**) [60], (**h**) [61], (**i**) [10], (**j**) [11], (**k**) [13], (**l**) [12], (**m**) [14], (**n**) [53], (**o**) [15], and (**p**) [17].

**Figure 3 polymers-17-00237-f003:**
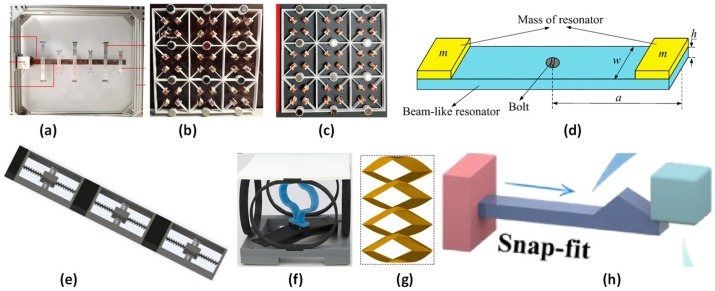
(**a**) Beam metastructure with double cantilever beam resonators [55]; (**b**) metamaterial structure with a local resonator having four angled cantilevers and a central magnet mass [56]; (**c**) representation of the basic unit cell of the layered metamaterial assembly [57]; (**d**) schematic diagram of the designed beam-like uncoupled force-moment type resonator [58]; (**e**) physical model of a periodic rod with high-static–low-dynamic stiffness (HSLDS) resonator [59]; (**f**) architected unit cell with three-spring configuration model [40]; (**g**) zigzag-base stacked-origami (ZBSO) metamaterial with tailored multistage stiffness [60]; and (**h**) polymeric snap-fit MM [61].

**Figure 4 polymers-17-00237-f004:**
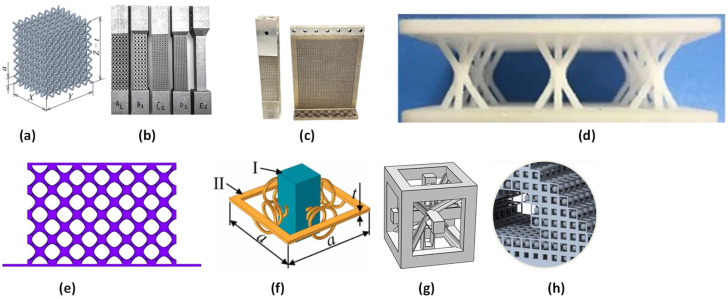
(**a**) Body-centered cube lattice structure model [10]; (**b**) different specifications of FBCCZ and full cross-section beam structures [11]; (**c**) beam- and blade-like FBCCZ and full cross-section structures [13]; (**d**) Kagome lattice plate [12]; (**e**) I-wrapped package (IWP-type) triply periodic minimal surface (TPMS) lattice structure [14]; (**f**) schematic diagram of the cell structure of vibration-damping metamaterial plates [53]; (**g**) SC-BCC-ABH lattice structure, simple cubic (SC) lattice, BCC support, and acoustic black hole (ABH) [15]; (**h**) unit cell cavity lattice with tuned mass dampers (TMDs) [17].

**Figure 5 polymers-17-00237-f005:**
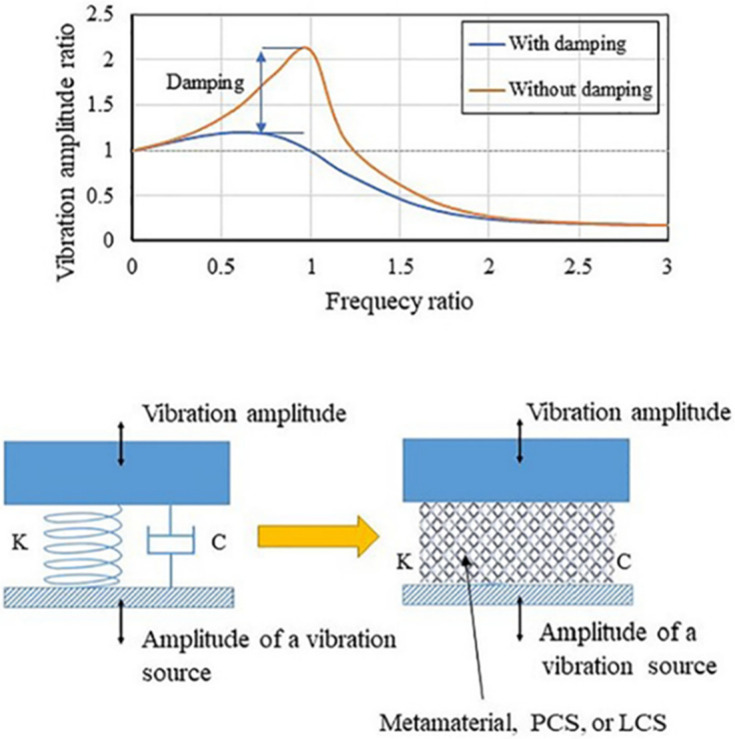
Vibration damping and isolation through LCS [75].

**Figure 6 polymers-17-00237-f006:**
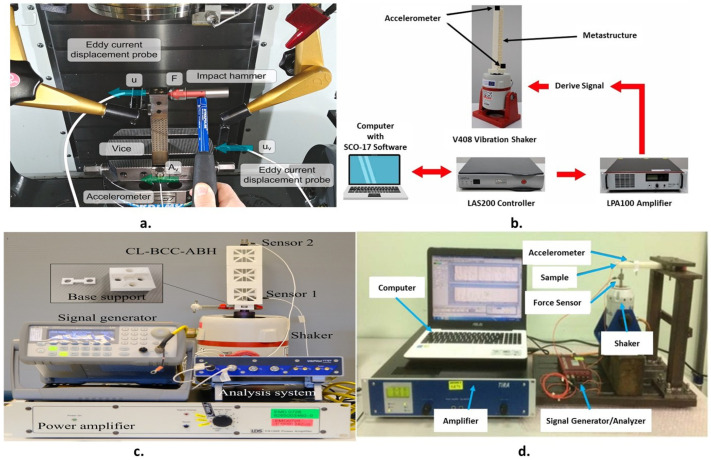
Different experimental setups used to characterize the damping of mechanical MM: (**a**) [11], (**b**) [111], (**c**) [15], and (**d**) [112].

**Figure 7 polymers-17-00237-f007:**
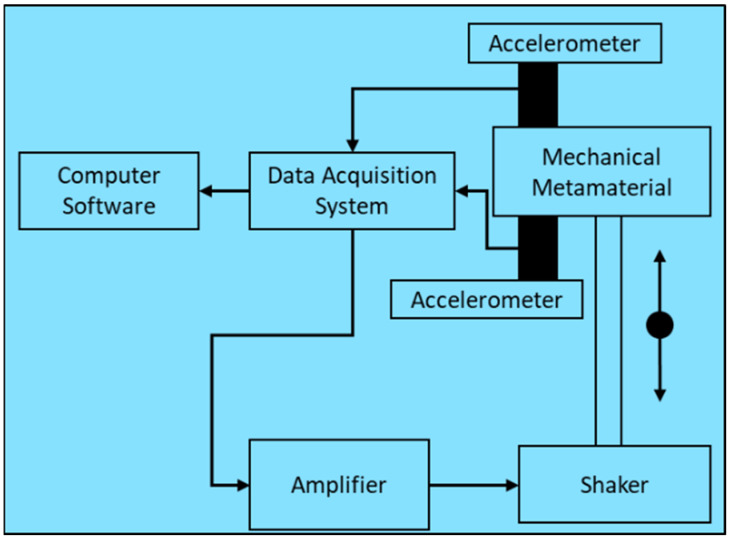
Layout of basic experimental facility for testing damping of MM *(Arrow denotes the direction of excitation of MM using shaker)*.

**Figure 8 polymers-17-00237-f008:**
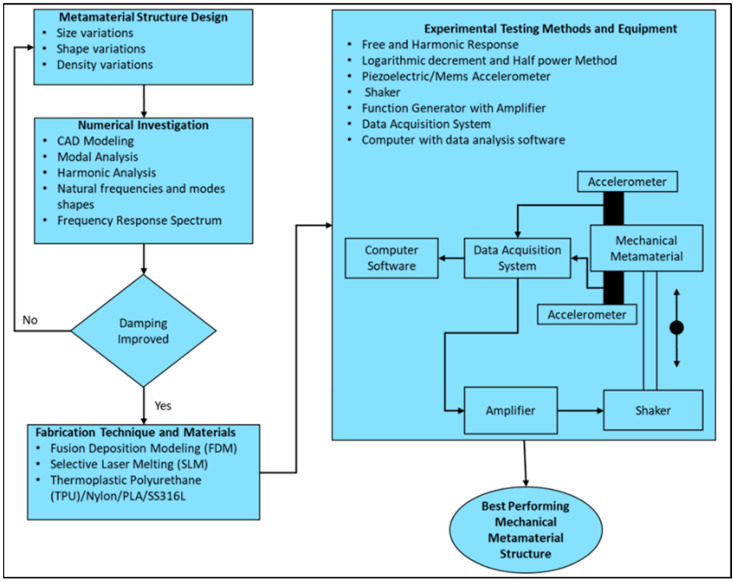
Proposed systematic procedure for future research.

**Table 1 polymers-17-00237-t001:** Review of the literature on MMs for bandgap formation, energy absorption for vibration suppression, attenuation, isolation, and shock absorption.

Reference	Structure(s)	Investigation/Applications	Key Findings
El-Borgi et al. [55]	Metamaterial beams	Bandgap formation	Observed two bandgaps from a numerical and experimental study
Ji et al. [67]	Metamaterials (auxetic, bandgap, and pentamode) and origami-based structures	Vibration mitigation and isolation	Reviewed advances in metamaterials and origami-based structures with future research directions
Anigbogu et al. [56]	Structure with periodic local resonators	Vibration attenuation and energy harvesting	Observed two bandgaps: 205–257 Hz and 587–639 Hz
Al Rifaie et al. [75]	Porous materials (PMs), periodic cellular structures (PCSs), and lattice cellular structures (LCSs)	Vibration isolation	Explored modeling, mechanical properties, and vibration techniques for PMs, PCSs, and LCSs with future proposed studies
Dalela et al. [65]	Star-shaped unit cell, HSLDS resonator, sinusoidal beam, and LR beam with X-shaped resonator	Stopband enhancement and broadening in varying frequency range	Explored MM development and focused on passive and active techniques to control vibration
Wang et al. [76]	Square cross-section with curved boundaries	Buildings’ seismic isolation and attenuation	Found that the proposed structure has excellent damping properties in a low-frequency band
Zhang et al. [77]	Stretch plate-based MM	Vibration isolation and energy absorption	Found that the metamaterial structure S8 is best for both vibration isolation and absorption
Anigbogu et al. [57]	Layered metamaterial structure	Frequency bandgap formation	Concluded that the local resonator’s length influences the shape and location of the local resonator frequency bandgap.
Hu et al. [71]	Internally coupled metamaterial	Energy harvesting and Vibration suppression in low-frequency	Observed that the new metamaterial model enhances energy harvesting and vibration suppression
Zhang et al. [72]	Pyramidal-truss metamaterial beam	Low-frequency vibration insulation	The position and bandgap width are affected by the resonator’s natural frequency, mass ratio, and the truss inclination angle; moreover, the proposed structure can obtain much broader bandgaps, with the same host beam structure, lattice constant, and resonator total masses in each unit cell
Lv et al. [78]	Finite lightweight locally resonant beam	Aerospace space-arm and antennas framework	The designed structure’s resonator mass and stiffness are more sensitive in achieving a broad-width frequency bandgap
Lv et al. [58]	Timoshenko beam with periodic 2-DOF uncoupled force-moment type resonators	Low-frequency range bandgaps	The study found an efficient approach to achieving broadband low-frequency bandgaps
Lv et al. [66]	L-joint LR beam	Vibration attenuation in lower-frequency range	Observed that the proposed structure can attenuate the propagation of the axial waves in the low-frequency range
Lv et al. [79]	Periodic 2-DOF force-type resonators	Vibration attenuation	Concluded that the resonator mass and stiffness affect the bandgap properties
Wang et al. [59]	Periodic rod with HSLDS resonator	Low-frequency bandgap	Creation of a very low-frequency bandgap for longitudinal waves propagating along the rod
Wu et al. [80]	Elastic metamaterial beam having X-shaped resonators	Bandgap adjustment and vibration control	The initial angle, length ratio, and layer number are important to manage the bandgap characteristics
Li et al. [81]	LR plate with multiple arrays of multi-DOF resonators	Broadband vibration suppression	The improved plane-wave expansion and extended plane-wave expansion approaches have outstanding efficiency and broad applicability for the proposed structure
Zhu et al. [82]	Chiral lattice elastic metamaterial beam	Vibration broadband suppression	The proposed structure’s experimental testing is performed to verify the design
Jian et al. [83]	Metamaterial-graded piezoelectric transducer beam	Vibration attenuation band broadening	Found that for the power spectral density of a random input, the excitation was 0.001 G2/Hz, and the RMS acceleration amplitude at the beam tip could be attenuated to 0.38 G
Fabro et al. [84]	Metamaterial and phononic crystals	Vibration suppression	Showed that the slowly changing method is appropriate to represent the ensemble statistics of bandgaps
Chen et al. [85]	Sinusoidally shaped lattice	Broadband vibration mitigation	Reported extreme Poisson’s ratio fluctuations between −0.7 and 0.5 over large tensile deformations of up to 50% for the proposed structure
Li et al. [86]	Sandwich-like metamaterial	Vibration attenuation and isolation	Showed that the stopband is affected by the resonator’s natural frequency and the mass ratio; the stopband width is mostly affected by the resonator’s damping ratio
Bukhari et al. [87]	Sliding mass metastructure	Wide-frequency-range vibration reduction	Showed that the resonator can adjust itself with external frequency once the slider achieves equilibrium position
Bae et al. [88]	Large beam structure	Vibration suppression	Showed that the present approach is effective in the vibration suppression of a large beam structure without the addition of significant weight
Chen et al. [89]	Cantilever beam	Vibration control	Found that the present structure with a tuned mass damper experiences 1.63 to 2.99 times the maximum vibration of that of an eddy current-tuned mass damper
Cheng et al. [90]	Flexible cantilever beams	Vibration suppression	Found that the magnet shunt damper can be used without any issue to reduce the vibration of flexible structures
Izard et al. [40]	Three-spring model	Energy dissipation	Observed that proposed materials show a very high Young’s modulus and damping combination, far better than those of the constituent phase
Correa et al. [73]	NS honeycomb structure	Mechanical energy dissipation	Found that the proposed structure may be modeled to dissipate mechanical energy comparable to traditional designs at relatively lower densities
Chen et al. [91]	Composite NS structure	Shock isolation and vibration control	Found from the impact tests that the proposed structure has good cushioning properties by tuning the acceleration threshold response and is reusable after snap-through behavior takes place
Ha et al. [74]	Cubic NS lattice structure	Energy absorption	Concluded that the proposed structure can absorb mechanical energy with full geometry recovery in all directions, and its energy absorption increases with its dimensions
Tan et al. [92]	Cylindrical NS structure	Shock isolation	Concluded from the impact tests that the cylindrical NS structure achieved good cushioning performance by adjusting the acceleration magnitude threshold after the snap-through behavior took place
Kovacic et al. [93]	Metastructure with integrated internal oscillators	Vibration attenuation	Observed that the proposed structure’s natural frequency is increased linearly and nonlinearly along the structure in line with new theoretical results
Xu et al. [61]	Polymeric snap-fit MM	Energy absorption	Found that the proposed structures achieve excellent impact resistance and energy absorption ability, so they can be assumed to be a suitable candidate for the development of shock absorbers
Tan et al. [94]	Pneumatically actuated tunable NS mechanical metamaterial	Vibration isolation and energy absorption	Mentioned that the multistage pattern transformation can be obtained through pneumatic actuation
Wen et al. [60]	ZBSO metamaterial	Energy absorption	Found that the proposed structure has many excellent advantages in comparison with the conventional mechanical metamaterials, i.e., material-independent, scale-invariant, and lightweight properties and excellent energy absorption capability

**Table 2 polymers-17-00237-t002:** Summary of the literature on vibration damping control of MMs.

Reference	Study Nature	Lattice Structure	Manufacturing Technique/Material
Monkova et al. [10]	Experimental work on vibration damping compression properties	BCC lattice structure	FDM technique/polymer acrylonitrile butadiene styrene (ABS)
Scalzo et al. [11]	Experimental and numerical work on damping properties	FBCCZ lattice structure	SLM technique/AISI 316L, AlSi10Mg
Wang et al. [12]	Experimental and numerical vibration and damping characteristics bandgap design	Kagome lattice structure	SLM technique/polymer nylon PA6, thermoplastic polyurethane as viscoelastic filling material
Scalzo et al. [13]	Experimental work on damping properties	FBCCZ Lattice Structure	SLM Technique/AISI 316L
Zhang et al. [14]	Experimental work on compression testing and dynamic vibration rate transfer testing	IWP TPMS lattice structure	SLM technique
Chu et al. [53]	Numerical, low-frequency damping	Rectangular sheet with conical spring	Lead and polymer nylon
Sheng et al. [15]	Simulations and experiments on high load-bearing capability and vibration suppression	SC-BCC-ABH lattice structure SC latticeBCC supportABH acoustic black hole	SLS technique/polymer PA12 nylon powder
Chen et al. [16]	Numerical and experimental work on low-frequency vibration suppression	NS metamaterial	SLS technique/polymer thermoplastic urethane
Janousek et al. [17]	Simulations of vibration reduction	Unit cell cavity lattice with TMD; a robotic arm was made from it	SLA resin material TOUGH1500
Simsek et al. [97]	Experimental work on compression testing and dynamic vibration rate transfer testing	Gyroid structures	Concept laser M2 (metal 3D-printing system) with high-temperature alloy named Haynes^®^ 188 (HS188)

## Data Availability

The data presented in this study are available upon request from the corresponding author.

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
