# Peer review of "Damping Optimization and Energy Absorption of Mechanical Metamaterials for Enhanced Vibration Control Applications: A Critical Review"

_polymers, 2025, doi:10.3390/polym17020237_

Round 1

Reviewer 1 Report

Comments and Suggestions for Authors

The manuscript entitled “A Critical Review of Mechanical Metamaterials for Vibration Control, Damping Optimization, and Energy Absorption Applications” presented a comprehensive review of the common properties of mechanical metamaterials, which is well organized and presented. It is of interest to readers, especially scholars and engineers working in the fields of impact mechanics. However, it also has some places that need to be revised. These are:

1. Re-write the title. The Vibration Control, Damping Optimization and Energy Absorption do not have similar characteristics, which is easy to make an assembly feeling.

2. In the abstract, “Their shapes are complex” is not appropriate because complex and simple are the relative concepts.

3. Fig.1 lacks an arrow pointing to the optical. The four negative phrases should be in a line, one after the other.

4. Vibration is a process in which a “mechanical structure undergo..., there is a typo symbol, i.e., .

5. For the negative Poissons ratio, the recent reference DOI: 10.1016/j.matdes.2024.113295 is suggested to include to compare their difference. For example, add to the “The energy absorption from vibration, shocks, impacts of the structures may be enhanced using negative stiffness structure configuration.”

6. Fig.2, the orders in the caption are given, but no labels in the image.

7. Equ. (1), please make the cos be normal.

8. Actually, auxetic metamaterials have excellent capabilities in energy absorption and vibration reduction. If this part can be added to the discussion, it will be better.

9. The biggest problem is that there are no obvious contents related to polymers, so please add some discussion about this in the introduction.

Author Response

Response to the Reviewer # 01 Comments:

Q-1: Re-write the title. The Vibration Control, Damping Optimization and Energy Absorption do not have similar characteristics, which is easy to make an assembly feeling.

Response: The authors are thankful for the valuable comment. The title is revised to avoid assembly feeling as follows: Damping Optimization and Energy Absorption of Mechanical Metamaterials for Enhanced Vibration Control Applications: A critical Review

It is highlighted in green color on page#01 in revised manuscript.

Q-2: In the abstract, “Their shapes are complex” is not appropriate because complex and simple are the relative concepts.

Response: The authors are thankful for the valuable comment. The complete sentence is modified to clarify the meaning of complex shapes. The modified sentence is highlighted in green color on page#1, lines 18-20 in the revised manuscript.

Q-3: Fig.1 lacks an arrow pointing to the optical. The four “negative” phrases should be in a line, one after the other.

Response: The authors are thankful for the valuable comment. The arrow is added pointing to optical in figure 1. The four “negative” phrases are also modified as recommended on page #2 in the revised manuscript.

Q-4: Vibration is a process in which a “mechanical structure undergo..., there is a typo symbol, i.e., ”.

Response: The authors are thankful for the valuable comment. The typo is removed, and the complete sentence is highlighted in green on page # 3, lines 100-102 in the revised manuscript.

Q-5: For the negative Poisson’s ratio, the recent reference DOI: 10.1016/j.matdes.2024.113295 is suggested to include to compare their difference. For example, add to the “The energy absorption from vibration, shocks, impacts of the structures may be enhanced using negative stiffness structure configuration.”

Response: The authors are thankful for the valuable comment. The recent reference is added as recommended to compare their difference and highlighted in green color on page #3, lines 119-121 in the revised manuscript.

The recent reference added is: [111] Zhao, Changfang, Jianlin Zhong, Hongxu Wang, Chen Liu, Ming Li, and Hao Liu. "Impact behaviour and protection performance of a CFRP NPR skeleton filled with aluminum foam." Materials & Design 246 (2024): 113295.

Q-6: Fig.2, the orders in the caption are given, but no labels in the image.

Response: The authors are thankful for the valuable comment. Fig. 2 is modified with image labels and figure caption is highlighted in green color on page #4 in the revised manuscript.

Q-7: Equ. (1), please make the “cos” be normal.

Response: The authors are thankful for the valuable comment. The “cos” is made normal as recommended and highlighted in green color on page#13 in equation (1) in the revised manuscript.

Q-8: Actually, auxetic metamaterials have excellent capabilities in energy absorption and vibration reduction. If this part can be added to the discussion, it will be better.

Response: The authors are thankful for the valuable comment. As suggested, in the revised version of the manuscript, the capabilities of the auxetic metamaterials for energy absorption and vibration reduction are discussed on page #6, lines 186-191, referenced [112-114] and are highlighted in green color.

The discussion is: Auxetic metamaterials are a category of mechanical metamaterials exhibiting negative Poisson's ratio. They possess excellent energy absorption and vibration damping capa-bilities in seismic and automotive applications [112-114].

The added references are: [112]         Sebaq, M. H., and Zishun Liu. "Energy absorption and vibration mitigation performances of novel 2D auxetic metamaterials." International Journal of Computational Materials Science and Engineering 13, no. 02 (2024): 2350022.

[113]   Xu, Chao, Qiwei Li, Lu Zhang, Qingping Liu, and Luquan Ren. "Glass sponge-inspired auxetic mechanical metamaterials for energy absorption." Journal of Bionic Engineering (2024): 1-17.

[114]   Saddek, Ahmed Abdalfatah, Tzu-Kang Lin, Wen-Kuei Chang, Chia-Han Chen, and Kuo-Chun Chang. "Metamaterials of auxetic geometry for seismic energy absorption." Materials 16, no. 15 (2023): 5499.

Q-9: The biggest problem is that there are no obvious contents related to polymers, so please add some discussion about this in the introduction.

Response: The authors are thankful for the valuable comment. It is also worth mentioning that many of studies presented in this review paper on metamaterials are making use of additive manufactured polymer materials, which justify and address the comment of the reviewer. Some of the references on polymer-based metamaterials are [10], [12], [15], [16], [17], [40], [47], [52], [58], [60], [102], [103], and [107]. These papers are highlighted in green color in the revised manuscript on page#19,20,21,22. In addition, the contents related to the polymer are mentioned in abstract lines 21-24, introduction, page#1, lines 47-48, page#2, lines 49-50, page #3, lines 85-87, and section 2, page#3, lines 124-125 and other sections throughout in the revised manuscript.

Reviewer 2 Report

Comments and Suggestions for Authors

The authors give a critical review of mechanical metamaterials for vibration control, damping optimization, and energy absorption applications. This work is interesting and has potential applications. I would like to recommend this manuscript for publication, after the following concerns are addressed properly.

[1] The part text in Figures 1 and 6 is relatively small. For clarity, please appropriately adjust the font size in these Figures.

[2] In this manuscript, many abbreviations were used. The abbreviations must be defined when they firstly appear. Especially, there is no need to repeat the definition of abbreviations. The abbreviation (AM) has been defined in the abstract part, so it is not necessary to define it again in the introduction part. Here, I only give an example. Please carefully check all abbreviations in the manuscript.

[3] The English of this manuscript needs careful examination. On page 4, “Fig. 3(d,e)” should be “Figs. 3(d, e)”. The space between two letters was missed. Here, I only give an example. Please carefully check similar expressions in the text of the manuscript to avoid similar errors.

[4] There is relevant work: Opt. Commun. 575, 131312 (2025). The paper can be cited to give audience a broader picture of this field.

Author Response

Response to the Reviewer # 02 Comments:

Q-1: The part text in Figures 1 and 6 is relatively small. For clarity, please appropriately adjust the font size in these Figures.

Response: The authors are thankful for the valuable comment. As suggested, in the revised version of the manuscript, the font size of the text in Figure 1 is adjusted on page# 2 and Figs. 6(a, b, c, d) on pag#15.

Q-2: In this manuscript, many abbreviations were used. The abbreviations must be defined when they firstly appear. Especially, there is no need to repeat the definition of abbreviations. The abbreviation (AM) has been defined in the abstract part, so it is not necessary to define it again in the introduction part. Here, I only give an example. Please carefully check all abbreviations in the manuscript.

Response: The authors are thankful for the valuable comment. As suggested, in the revised version of the manuscript, all abbreviations are defined once, and repetition is avoided. The corrections are highlighted in yellow color throughout the revised manuscript.

Q-3: The English of this manuscript needs careful examination. On page 4, “Fig. 3(d,e)” should be “Figs. 3(d, e)”. The space between two letters was missed. Here, I only give an example. Please carefully check similar expressions in the text of the manuscript to avoid similar errors.

Response: The authors are thankful for the valuable comment. As suggested, in the revised version of the manuscript, corrections have been made for all the figures numbers in the text and are highlighted in yellow color.

Q-4: There is relevant work: Opt. Commun. 575, 131312 (2025). The paper can be cited to give audience a broader picture of this field.

Response: The authors are thankful for the valuable comment. As suggested, in the revised version of the manuscript, the paper is added in the introduction section on page#2, line 62 and is highlighted in yellow color.

The added reference is: [110] Yu, Lili, Fan Ji, Tian Guo, Zhendong Yan, Zhong Huang, Juan Deng, and Chaojun Tang. "Ultraviolet thermally tunable silicon magnetic plasmon induced transparency." Optics Communications 575 (2025): 131312.

Reviewer 3 Report

Comments and Suggestions for Authors

The paper worth to be published but needs serious improvements. In the attached file several observations and suggestions are done. Probably some of them may be missing as not noticed, so a complete and careful revision of the paper should be done. The most important observations are mentioned hereby:

a) write everywhere Poisson (as last name) and not poisson (a fish in French);

b) give everywhere full name before using an acronym;

c) in Fig. 2 add letters for figures mentioned in legend;

d) page 4 line 138 -  what is a "lattice constant"?

e) page 5 line 161 - what is a "bi function"?

f) page 5 line 168 - to which AI methods do you refer?

g) for Table 1 - make some comments on this table in the text of the paper;

h) refer to Figure 4 before it in the text;

i) a general comment - all variables have to written in text as italics, as they appear in the equations;

k) Figure 6 - figures are not clear; improve quality of pictures and written text;

l) for References - write name of the authors in a consistent manner as being the same sequence everywhere; now there are three different ways in which names are written; name of journals should be written in a similar manner; year of publication written in the same manner.

In general, the manuscript has to be carefully checked for scientific soundness explanations and used English language.

Comments on the Quality of English Language

Text should be checked by a professional English speaker. In many places text is confusing, and should be rephrased, and some mistakes are done.

Author Response

Response to the Reviewer # 03 Comments:

Q-1: write everywhere Poisson (as last name) and not poisson (a fish in French);

Response: The authors are thankful for the valuable comment. As suggested, in the revised version of the manuscript, the word Poisson is corrected and highlighted in turquoise color on pages # 2, 3, 6 and 8.

Q-2: give everywhere full name before using an acronym;

Response: The authors are thankful for the valuable comment. As suggested, throughout the revised manuscript, full names are used before the acronyms and are highlighted in turquoise color.

Q-3: in Fig. 2 add letters for figures mentioned in legend;

Response: The authors are thankful for the valuable comment. As suggested, in the revised version of the manuscript, letters are added in the legend of Fig. 2 on page #4.

Q-4: page 4 line 138 -  what is a "lattice constant"?

Response: The authors are thankful for the valuable comment.

It is defined as follows: The lattice constant refers to the repeating unit length of the metamaterial's structure.

It is now defined on page#4, lines 140-141 of the revised manuscript and highlighted in turquoise color.

Q-5: page 5 line 161 - what is a "bi function"?

Response: The authors are thankful for the valuable comment.

The word “bi function” means: Dual function of vibration attenuation and energy harvesting.

The word “bi” is replaced by “dual”, explanation is added to it in parentheses to clarify its meaning and highlighted in turquoise color on page# 5, line 162, in the revised manuscript.  

Q-6: page 5 line 168 - to which AI methods do you refer?

Response: The authors are thankful for the valuable comment.

AI methods refer to: Generative design and topology optimization. A sentence is added to briefly mention the AI methods and highlighted in turquoise color on page#5, lines 169-171 of the revised manuscript.

Q-7: for Table 1 - make some comments on this table in the text of the paper;

Response: The authors are thankful for the valuable comment. As suggested, in the revised version of the manuscript, Table 1 is discussed briefly in the text on page#6, lines 204-207, 215-217 and highlighted in turquoise color.

Q-8:  refer to Figure 4 before it in the text;

Response: The authors are thankful for the valuable comment. As suggested, in the revised version of the manuscript, the Figure is now referred before it is placed in the manuscript and is highlighted in turquoise color on pages#10 and 11.

Q-9: a general comment - all variables have to written in text as italics, as they appear in the equations;

Response: The authors are thankful for the valuable comment. As suggested, in the revised version of the manuscript, all variables are written in text as italics and highlighted in turquoise color on pages#13 and 14.

Q-10: Figure 6 - figures are not clear; improve quality of pictures and written text;

Response: The authors are thankful for the valuable comment. As suggested, in the revised version of the manuscript, the quality of the pictures and text is improved in Figure 6 on pag#15.

Q-11: for References - write name of the authors in a consistent manner as being the same sequence everywhere; now there are three different ways in which names are written; name of journals should be written in a similar manner; year of publication written in the same manner.

Response: The authors are thankful for the valuable comment. As suggested, in the revised version of the manuscript, all references are modified to a uniform style to make the names of authors, journals and publication years consistent.

Q-12: In general, the manuscript has to be carefully checked for scientific soundness explanations and used English language.

Response: The authors are thankful for the valuable comment. The comments provided in the attached pdf file are implemented in the revised manuscript and are highlighted in turquoise color.

Round 2

Reviewer 1 Report

Comments and Suggestions for Authors

The authors have addressed my comments well. It is recommended to published it.

Reviewer 3 Report

Comments and Suggestions for Authors

Names of authors are still not written in a consistent manner, as following the same pattern. Apply exactly the instructions given by the Editor.

However, with these corrections and taking into account the responses given to reviewers, the paper can be published.